# Genome-Wide Selective Signatures Reveal Candidate Genes Associated with Hair Follicle Development and Wool Shedding in Sheep

**DOI:** 10.3390/genes12121924

**Published:** 2021-11-29

**Authors:** Zhihui Lei, Weibo Sun, Tingting Guo, Jianye Li, Shaohua Zhu, Zengkui Lu, Guoyan Qiao, Mei Han, Hongchang Zhao, Bohui Yang, Liping Zhang, Jianbin Liu, Chao Yuan, Yaojing Yue

**Affiliations:** 1Lanzhou Institute of Husbandry and Pharmaceutical Sciences, Chinese Academy of Agricultural Sciences, Lanzhou 730050, China; lzh887246@163.com (Z.L.); swb887246@126.com (W.S.); guotingting@caas.cn (T.G.); lijianye218@163.com (J.L.); luzengkui@caas.cn (Z.L.); qhqiaogy@163.com (G.Q.); 82101186189@caas.cn (M.H.); 18837101296@163.com (H.Z.); yangbh2004@163.com (B.Y.); liujianbin@caas.cn (J.L.); yuanchao@caas.cn (C.Y.); 2College of Animal Science and Technology, Gansu Agricultural University, Lanzhou 730070, China; zhu87932890@126.com (S.Z.); zhanglp512@163.com (L.Z.)

**Keywords:** sheep, SNP chip, selective signatures, Fst, θπ ratio, XP-EHH

## Abstract

Hair follicle development and wool shedding in sheep are poorly understood. This study investigated the population structures and genetic differences between sheep with different wool types to identify candidate genes related to these traits. We used Illumina ovine SNP 50K chip genotyping data of 795 sheep populations comprising 27 breeds with two wool types, measuring the population differentiation index (Fst), nucleotide diversity (θπ ratio), and extended haplotype homozygosity among populations (XP-EHH) to detect the selective signatures of hair sheep and fine-wool sheep. The top 5% of the Fst and θπ ratio values, and values of XP-EHH < −2 were considered strongly selected SNP sites. Annotation showed that the *PRX*, *SOX18*, *TGM3*, and *TCF3* genes related to hair follicle development and wool shedding were strongly selected. Our results indicated that these methods identified important genes related to hair follicle formation, epidermal differentiation, and hair follicle stem cell development, and provide a meaningful reference for further study on the molecular mechanisms of economically important traits in sheep.

## 1. Introduction

Hair sheep is a kind of sheep with wool shedding that can adapt to a variety of climatic conditions and ecological environment. Hair sheep have coats with few woolly fibers or fibers that are shed annually. Not only do they not have to shear their wool often, but they are more resistant to parasites and diseases. While all sheep technically have both wool and hair fibers, hair sheep breeds have higher proportions of the latter. Nevertheless, since the breeds are so adaptable, they can thrive in cold climates, too, as they grow thicker undercoats when they move further north [1]. Due to the obvious depilation characters and high anti-parasite [2], hair sheep are becoming popular in the United States and other temperate regions, such as the Dorper, Wiltshire Horn, Katahdin sheep, and Barbados Black belly. The Dorper sheep is a heavy muscled sheep breed that originated in South Africa. It is a result of the breeding between Dorset Horn rams with Blackhead Persian ewes [3]. Dorper sheep breed has two distinct types the White Dorper and the traditional Dorper. The White Dorper is as its name implies a pure white-haired sheep and the traditional Dorper has a white body and black head much lie the Blackhead Persian breed. They are one of the most popular breeds of sheep in the United States today due to their shedding characteristics and superior conformation. The Barbados Black belly sheep is a breed originated in the Caribbean islands and is famous for its parasite resistance and cold tolerance [1]. Compared with wool, mutton has become the main economic source of farmer, furthermore, the labor cost required for shearing in the feeding process is higher than the value of wool itself. Therefore, in order to reduce breeding costs, people have a strong interest in sheep varieties with natural depilation [4,5]. The purpose is to reduce the feeding cost and increase the economic income of sheep on the basis of maintaining superior characteristics, such as growth rate, fertility and disease resistance. Compared with pure-bred hair sheep, their hybrid offspring can grow more wool fibers but still have the traits of natural shedding [6,7]. Although the main factors affecting wool shedding in this sheep are not clear, there is preliminary evidence that the shedding is strongly controlled by heredity, most of its expression is a single dominant autosomal gene [5]. Therefore, it would be of interest to explore the genetic mechanisms of wool shedding using selective signatures to identify genes related to sheep hair follicle development and the control of wool shedding and to understand their functions.

With the development of high-throughput phenotyping technology and molecular markers, the use of selective signatures can identify the imprints left in the genome during selection [8]. Among many selective signatures analysis methods, the most commonly used include Fst based on the population differentiation index. When the difference in the frequency of the same allele in different populations is greater than the expectation that the two populations are in neutral conditions, the population differentiation method can reject the neutral hypothesis and infer the presence of a selective effect. The Fst methods generally adopt the genome unit point scanning strategy. Fst can detect differences in allele frequencies between populations and effectively detect the selected site in the population [9]. The θπ ratio based on gene heterozygosity is also a common detection method. This is a parameter that measures the polymorphism of a specific population. The larger the π value, the higher the polymorphism of its corresponding subgroup [10]. Li et al. [11] performed selection signal analysis (Fst and θπ ratio) on resequencing data from 70 cashmere and 14 non-cashmere goats, screening out several genes potentially involved in cashmere fiber formation fibroblast growthfactor 5(*FGF5*), serine/threonine-protein kinase (*SGK3*), lin-like growth factor binding protein 7(*IGFBP7*), oxytocin receptor(*OXTR*), and rho associated coiled-coil containing protein kinase 1(*ROCK1*).

XP-EHH is an extension of the statistical principles of EHH and His [12] and can be used to detect genome-wide selective roles in a given population. Zhong et al. [8] used the EHH method to detect the important core regions of 202 Jinhua pigs, and screened the genes related to economic traits such as meat quality, reproductive ability, immune response, and external traits, identifying the microRNA ssc-mir-365-2 and the lysine demethylase 8(*KDM**8*), rabaptin, RAB GTPase binding effector protein 2(*RABEP**2*), GSG1 like(*GSG**1L*), ras homolog, mTORC1 binding(*RHEB*), and rabphilin 3A like(*RPH3AL*) genes. Yin et al. [13] performed a genome-wide sequencing of six Pengxian yellow chickens and, through selective scanning analysis, detected several regions with strong selection signals, including 497 protein-coding genes. These genes were involved in developmental processes, metabolic processes, responses to external stimuli, as well as other biological processes, such as digestion ATP binding cassette subfamily G member 5(*ABCG**5*), ATP binding cassette subfamily G member 8(*ABCG**8*) and adrenoceptor β 1(*ADRB**1*), muscle development and growth sphingomyelin phosphodiesterase 3(*SMPD**3*), neural EGFL like 1(*NELL**1*) and BicC family RNA binding protein 1(*BICC**1*), and decreased immune function metastasis associated 1 family member 3(*MTA**3*). Zhang et al. [14] successfully detected the whole-genome copy number variation (CNV) of 318 individuals from 24 Chinese native cattle breeds and 37 yaks as outgroups by analyzing the population structure and adaptability to high altitude. Although selective signatures have been successfully used to screen genes related to livestock economic traits, data are lacking about the sheep wool shedding. Here, we selected 27 sheep breeds in populations of two wool types and, using Illumina ovine SNP 50K chip typing data to analyze the selective signatures. Based on the Fst, θπ ratio, and XP-EHH results to screen candidate genes related to sheep hair follicle development and depilation traits, in order to provide a reference for sheep molecular breeding.

## 2. Materials and Methods

### 2.1. Ethics Statement

This study was conducted according to the guidelines for the care and use of laboratory animals promulgated by the State Council of China. This research was approved by the Animal Management and Ethics Committee of Lanzhou Institute of Animal Husbandry and Veterinary Medicine, Chinese Academy of Agricultural Sciences (license number: 2019-008).

### 2.2. Sample and Data Processing

In this study, a total of 795 individuals from 27 breeds of domestic and foreign sheep were selected for inclusion. The sheep were divided into fine-wool sheep and hair sheep, according to the wool type. The names, abbreviations, and sample size information for each variety are shown in Table 1. The Illumina ovine SNP 50K chip genotyping data for the 27 breeds used in the study were obtained from the International Sheep Genomics Consortium (ISGC) (https://www.sheephapmap.org/, accessed on 13 June 2021), National Animal Genome Research Program (NRSP) (https://www.animalgenome.org/sheep/community/, accessed on 25 June 2021), a Web-Interfaced next generation Database dedicated to genetic Diversity Exploration (WIDDE) (http://widde.toulouse.inra.fr/, accessed on 9 July 2021), and the Ontario Sheep Farmers (OSF) (https://www.ontariosheep.org/, accessed on 21 July 2021). Because the 50K chip genotyping data in the databases uses different versions of the sheep reference genome, the SNP location information was reordered using the locations of the Ovis_aries_v4.0 reference genome. Quality control was performed with Plink (v1.90) [15] software. The quality control standard [16] was as follows: (1) The genotype information of the autosomal chromosomes 1–26 was extracted for subsequent analysis, with the parameter—chr1-26; (2) Sites with SNP deletion rates >0.1 were excluded, with the parameter—geno 0.1; (3) The minor allele frequency (MAF) was >0.01, with the parameter—maf 0.01. Finally, 37 895 SNPs were obtained for further analysis.

### 2.3. Population Structure

Principal Component Analysis (PCA) is a dimensionality reduction statistical method aided by orthogonal transformation that recombines the original variables into a new set of independent comprehensive variables [17]. In population genetics, samples are clustered according to their principal components based on the degree of SNP differences in the individual genome and the characteristics of each trait. In the present study, PCA was conducted by using Plink (v1.90) software with the parameter plink—pca 3. The results were visually displayed using the ggplot package in R [18].

### 2.4. Selective Signatures

The combined methods of the Fst unit point, sliding window, and θπ ratio were used to analyze the selective signatures of sheep populations with different wool types. Data analysis was completed using Vcftools (0.1.15) [19] software. In brief, the Fst value in the chromosome was calculated with 500 kb as the sliding window and 50 kb as the step window [20,21], followed by extraction of the sites in the window where the Fst and θπ ratio were in the top 5% [22]. These sites were regarded as significant SNP candidate sites for the selective signatures. As the degree of linkage disequilibrium between sites is gradually reduced due to increasing marker spacing, the EHH of different lengths caused by selection can be observed in the genome. Although EHH can accurately select relevant genes under selection pressure, it does not distinguish whether these genes are newly mutated alleles or ancestral alleles [22]. Therefore, this study used the integrated haplotype score (iHS) test and the cross-population extended haplotype homozygosity (XP-EHH) test. The rehh package in R [23] was used for intra-group and inter-group selection signal detection. The ihh2ihs command in rehh was used to calculate the iHS value of the sheep population, after which the ies2xpehh command in rehh was used to calculate the XP-EHH value of the hair sheep, to detect the sites with different selection pressures between the hair sheep and fine-wool sheep [24].

### 2.5. Candidate Gene Detection and Annotation

The relevant SNP information was obtained after the detection of the selection pressure sites for the hair sheep population. Regions 50 kb upstream and downstream of the SNP are considered as the selected regions of signal action [25], with reference to the Ovis_aries_v4.0 genome (https://www.ncbi.nlm.nih.gov/assembly/GCF_000298735.2, accessed on 3 August 2021) information annotating the regions of selective signatures.

## 3. Results and Data Analyses

### 3.1. Analysis of the Population Structure

The PCA of the genotyping data from the 27 breeds is shown in Figure 1. PC1 could explain 4.42% of the genetic variation, while PC2 could explain 1.74%. Through PC1, PC2, and PC3, it was found that the sheep populations clustered together according to wool types, indicating that the genetic relationship between the two wool types is relatively distant.

### 3.2. Fst and θπ Ratio

Compared with the fine-wool sheep, shedding is characteristic of the hair sheep. It has obvious shedding characteristic and high anti-parasite. Although the main factors affecting wool shedding in this sheep are not clear, there is preliminary evidence that the shedding is strongly controlled by heredity, most of its expression is a single dominant autosomal gene. Through the detection and analysis of selective signatures between two wool-type sheep populations, the genomic regions affecting shedding in the hair sheep were identified. Fst and θπ ratio analysis of the hair sheep were referenced to the fine-wool sheep using Plink (v1.90) software to analyze the 37,895 SNP sites to obtain the pairwise genetic differentiation index Fst and the θπ ratio between two wool-type populations. At the genome-wide level, the top 5% SNP sites identified by the Fst (Fst > 0.0536) and θπ ratio (θπ ratio > 1.1398) analysis were taken as candidate target sites. Among them, 1262 SNP sites were higher than the threshold line (Figure 2), including 110 sites at the tail of the Fst distribution (Fst > 0.25) and 389 sites with large deviations (θπ ratio > 2).

### 3.3. XP-EHH Analysis

By comparing the differences in extended haplotype homozygosity between the two wool-type sheep populations, the genomic regions affecting shedding in the hair sheep were examined. The fine-wool sheep population was used as the reference population and the hair sheep as the experimental population for selection signal analysis. As shown in Figure 3, the sites with XP-EHH < −2 were selected as SNP sites in the hair sheep population [26]. In addition, a total of 618 SNP sites were selected in the hair sheep population. The frequency distribution histogram (Figure 4) of the XP-EHH values of hair sheep showed that the XP-EHH between the hair and fine-wool sheep followed a normal distribution.

### 3.4. Gene Mapping and Functional Annotation

With reference to the sheep Ovis_aries_v4.0 genome information, the Fst, θπ ratio, and XP-EHH were used to screen the selected SNP sites in the hair sheep population for annotation. Of these, 1262 were identified in the combined Fst and θπ ratio analyses, and 618 sites in the XP-EHH analysis. Next, a Venn diagram was drawn to illustrate these results, which showed 115 SNP loci in the overlapping region (Figure 5). Interestingly, after gene mapping and functional annotation, a G→T mutation was detected at SNP site 41339449 in Chr5 at a distance of 2996 bp from *TCF**3*, a C→T mutation was detected at SNP site 52044015 in Chr13 at a distance of 825 bp from *TGM*3, a G→A mutation was seen at SNP site 52055150 located within *TGM*3, a C→T mutation was observed at SNP site 53089456 in Chr13 at distance of 26,609 bp from *SOX**18*, and a G→A mutation was detected at SNP site 48786396 in Chr14 located within *PRX* (Table 2). Deep analysis of the two SNPs sites, 13-52055150 and 14-48786396, were performed using MEGA (5.0) and BioEdte (7.0) software. The results showed that SNP site 52055150 in TGM3 was located in the non-coding region between exon 8 (52051614–52051922) and exon 9 (52055639–52055881), which does not participate in the transcription process and does not cause changes in amino acids and proteins. The SNP locus 14-48786396 within the PRX gene is in exon 7 and is involved in coding amino acid at position 387. This site changes the codon from AGC to AAC and is a missense mutation, resulting in the change of its encoded amino acid from aspartic acid (N) to serine (S). A single mutation at this site causes changes in its mRNA secondary structure, in turn affecting the structure of the protein.

## 4. Discussion

In the current study, we selected two sheep populations with different wool types to investigate the clustering and genetic relationships of the selected samples through PCA. PCA is an unsupervised linear technique for dimension reduction and allows the extraction of axes of maximal variation from datasets [27]. This method was used to allocate them to groups using genotypic markers. When the classification is not obvious, and animals of the same breed tend to be located close together in PCA plots [28]. As it was assumed that some of the animals used in the current study may not have been pure-bred and unmanaged crossbreeding could have occurred in their herds, PCA was used to allocate them to groups using genotypic markers [29]. The results showed that the PC1 separated the 27 sheep breeds according to different wool types, including 470 fine-wool sheep and 325 hair sheep, with the clear separation between the fine-wool and hair sheep indicative of a relatively distant genetic relationship between the two populations. In contrast, PC2 and PC3 were distinct from each other in terms of the different varieties but the distance was small. In recent years, the analysis of selective signatures has become an important tool for screening the economic traits of animals and plants. Fst can scan for genome-wide SNP sites, calculating the Fst value for each SNP, using a range of values from 0 to 1, where 0 represents no differentiation between the populations at any of the sites and 1 represents complete differentiation [30]. Fst values between 0.05–0.15 indicate moderate genetic differentiation between populations, while values of 0.15–0.25 suggest a large difference, with values over 0.25 indicative of substantial differentiation [30,31]. To improve the detection accuracy, the combination of unit point and sliding window is used to reduce the probability of false positives [32]. The θπ ratio is analyzed according to gene heterozygosity, and the ratio between the two populations is then calculated. The higher the degree of genome selection, the more the θπ ratio deviates from 1 [11]. XP-EHH is based on the principle of gene linkage disequilibrium. After manual selection and genetic improvement, a large amount of chromosomal recombination is apparent in a population. However, the presence of linkage results in the generation-to-generation transmission of neutral sites near the mutant gene, forming a long range of haplotype homozygosity on the chromosome. Compared with the Fst that can only calculate the site of differentiation, XP-EHH statistics can further identify the population where the selection occurred. We used a combination of these methods together with mutual validation to improve the accuracy of gene localization, resulting in the identification of 115 SNP sites identified in the overlapping region between the three methods; of these, gene functional annotation yielded 106 significant genes, including 64 with known functions.

Based on the overlapping region, we identified four potential genes (*PRX*, *SOX**18*, *TGM**3*, and *TCF**3*) that were directly or indirectly associated with hair follicle development and wool shedding. Marker 14_48786396 on Chr14, associated with the skin and hair follicles, was located within *PRX*. Previous studies have demonstrated that members of the *PRX* gene family encode peroxidases, which can utilize the activated cysteine to remove peroxides and hydroxyl radicals. The *PRX* homotype (I–VI) is related to cell proliferation, differentiation, and anti-apoptosis [33]. In terms of skin diseases, *PRX*II was found to be expressed in the vascular endothelial cells of normal skin [34]. In addition, in previous studies on the isoform-specific expression of peroxidases in rat epidermis and hair follicles, three *PRX* isoforms were identified, with PrxII located in the epidermal cells and epidermal accessory structures of the dermis (hair follicles, eccrine glands, and sebaceous glands) [35]. Primary cilia were observed in the hair follicle dermal papillae of Prx1-Cre skin in mice expressing Prx1-driven Cre recombinase [36]. These findings suggest that *PRX* represents a highly significant gene in the determination of wool properties.

The *SOX* gene family encodes transcription factors. Its products have a HMG motif conserved region [37], that is involved in the regulation of embryonic development and the determination of cell fate. The encoded proteins act as transcriptional regulators in the development of hair, blood vessels, and lymphatic vessels through complexation with other proteins. *SOX18* belongs to the SRY-related HMG domain family and is one of the important members of the SOX transcription factor F (SOX F) subfamily. It regulates angiogenesis [38], lymphangiogenesis [39], and hair follicle differentiation [40]. It is located in Chr20 and Chr2 in human and mouse, respectively, and plays an important role in the development of blood vessels and lymphatics [41]. It has been found that *SOX18* is expressed in hair follicles during mouse embryonic development [40], and a *SOX18* point mutation has been found in two different mutant mouse alleles. Compared with the wild-type, the fusion protein containing this mutation lacked the ability to activate transcription and it has been found that the mutation induces hair follicle defects in spontaneous mouse mutants [41], indicating that *SOX18* plays a vital role in hair follicle development or function. Previous studies on the effects of *SOX18* in endothelial cells found that the gene was transiently expressed in the mesenchyme of endothelial cells and hair follicles during mouse embryogenesis and played a key role in the development of hair follicles [42]. In the induction experiment of dermal papilla on hair follicle growth stage, it was found that *SOX18* was specifically expressed in the dermal papillae of all dorsal hair follicle types [43]. A study on human patients and their offspring identified a recessive mutation in *SOX18* that played a key role in the development of hair, blood vessels, and lymphatic vessels [44], and was mainly manifested clinically as sparse hair-lymphedema-telangiectasia [39]. In summary, the abnormal expression of *SOX18* can cause sparse hair-lymphedema-telangiectasia syndrome, and *SOX18* has been identified as a key gene in hair follicle formation. The SNP locus located at 53089456 was detected on chr13, at a 26,609 bp distance from the *SOX18* locus and was found to be strongly selected in the hair sheep population. Therefore, it is speculated that *SOX18* plays a key role in sheep hair follicle formation. Although the abnormal expression of *SOX18* may be related to sheep hair follicle development and depilation traits, this requires further verification.

Marker 13_52044015 and 13_52055150 on Chr13, were located beside (825 bp) and within *TGM*3, which encodes type 3 glutamine transaminase (tgase-3) [45], an enzyme with Ca^2+^-dependent transaminase activity in epidermal non-proliferative layers and hair follicles [46]. Mutation of human *TGM**1* leads to type I autosomal recessive lamellar ichthyosis, which is characterized by skin surface squamous epidermal hyperplasia and barrier defects [47]. On the other hand, *TGM**3* knockdown in mice resulted in developmental delays in skin barrier formation in utero, with abnormalities in hair follicle function [48]. It is thus inferred that *TGM**3* is closely related to the development of hair follicles, although its mechanism of action requires further research.

Hair follicle stem cells are mainly located in the bulge of the outer root sheath of the hair follicles. They belong to the adult stem cell, which can differentiate into epidermis, hair follicles, and sebaceous glands, and participate in the process of skin wound healing [49]. For example, in human skin, hair follicle pluripotent stem cells are considered to be necessary for the hair cycle and epidermal wound repair [50], and *TCF**3* is naturally expressed in hair follicle pluripotent stem cells. During the growth phase of the hair cycle, *TCF**3* is expressed not only in quiescent pluripotent stem cells but also in the outer basement layer of the newly formed outer root sheath [51]. A study found the Tcf protein not only plays a role in Wnt signaling, but also acts as an inhibitor when β-catenin is low or absent. Moreover, Tcf3 and Tcf4 may have wnt-dependent and wnt-independent effects, which are essential for the establishment and maintenance of all skin epithelial stem cells [52]. When the level of Tcf3/4-Tle is high, hair follicle stem cells can maintain their characteristics, but still static. When Tcf3/4-Tle levels drop or Wnt-β-catenin levels rise, this balance changes and hair regeneration begins [53]. In addition, the *TCF**3* suppressor gene includes transcription regulators of the epidermal, sebaceous gland, and hair follicle differentiation program [50], regulates cell proliferation and apoptosis, induces hair follicle stem cell differentiation, and promotes the transformation of hair follicles from the resting phase to the growth phase [54]. These results suggest that *TCF**3* plays a vital role in hair follicle stem cells and the Wnt pathway and its associated signal transduction.

## 5. Conclusions

The current study used Fst, the θπ ratio, and XP-EHH to detect the selective signatures of sheep SNP chip typing data. A total of 115 SNPs sites were identified, of which the *PRX*, *SOX**18*, *TGM**3*, and *TCF**3* genes were found to be closely related to hair follicle formation, epidermal differentiation, hair follicle stem cell development, and wool shedding, located on Chr5, Chr13, and Chr14. These findings provide a reference for the study of the mechanism of sheep hair follicle development. In future studies, we will use high-density chips and an increased number of populations to study these genes and their functions in hair development in greater depth.

## Figures and Tables

**Figure 1 genes-12-01924-f001:**
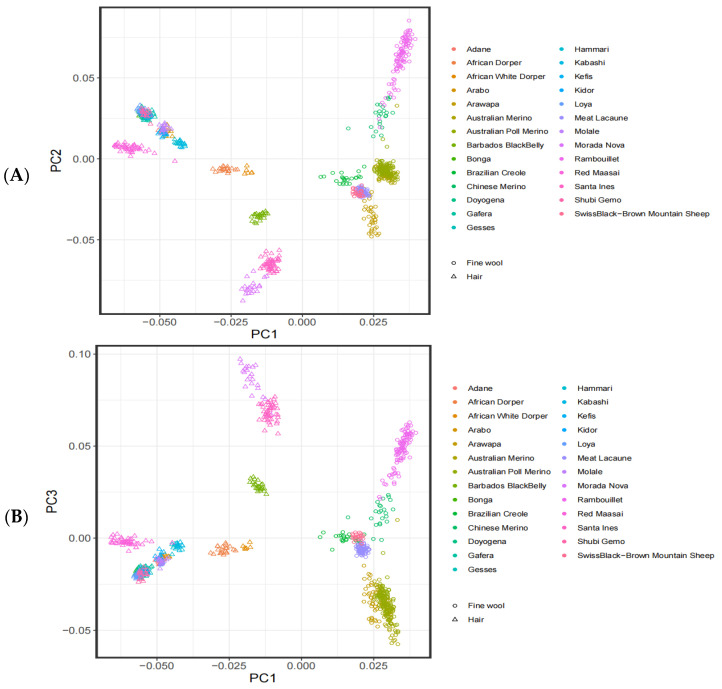
Principal component analysis of the different sheep populations. (**A**) Analysis results of PC1 and PC2 between the fine-wool sheep and hair sheep populations; (**B**) Analysis results of PC1 and PC3 between the fine-wool sheep and hair sheep populations.

**Figure 2 genes-12-01924-f002:**
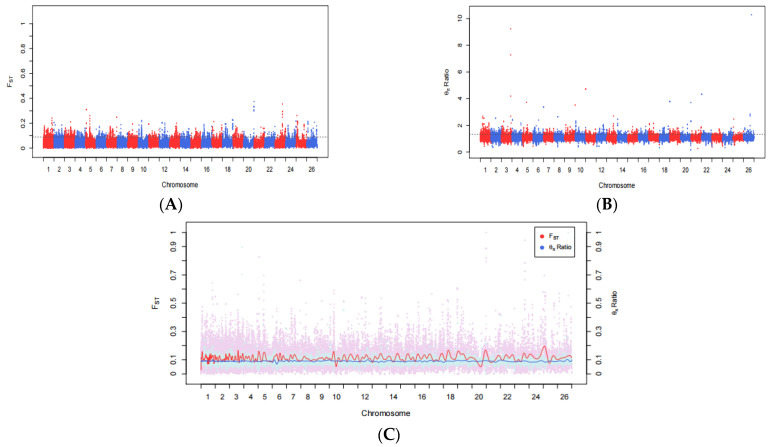
Analysis of selection signal distribution on sheep autosomes. (**A**) Manhattan map of Fst; (**B**) Manhattan map of θπ ratio; (**C**) Manhattan map of Fst and θπ ratio joint analysis.

**Figure 3 genes-12-01924-f003:**
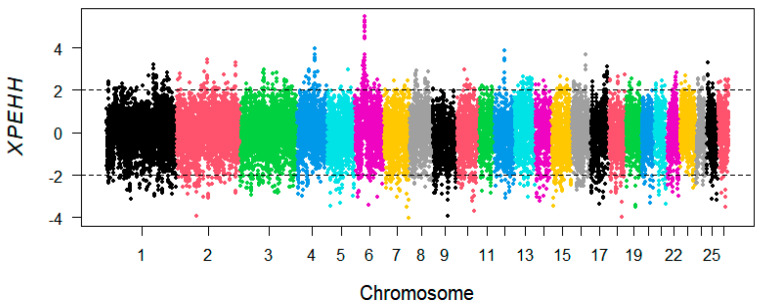
Analysis of selection signal distribution on sheep autosomes. Chromosomes 1 to 26 are shown in different colors; the horizontal dotted black line indicates the genome-wide significance level.

**Figure 4 genes-12-01924-f004:**
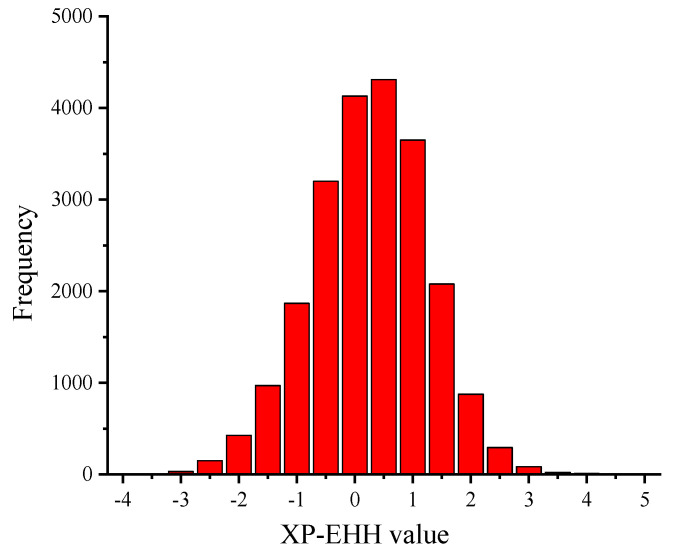
The distribution of XP-EHH in hair sheep and fine-wool sheep. The abscissa represents the haplotype value and the ordinate represents the number of occurrences of each region.

**Figure 5 genes-12-01924-f005:**
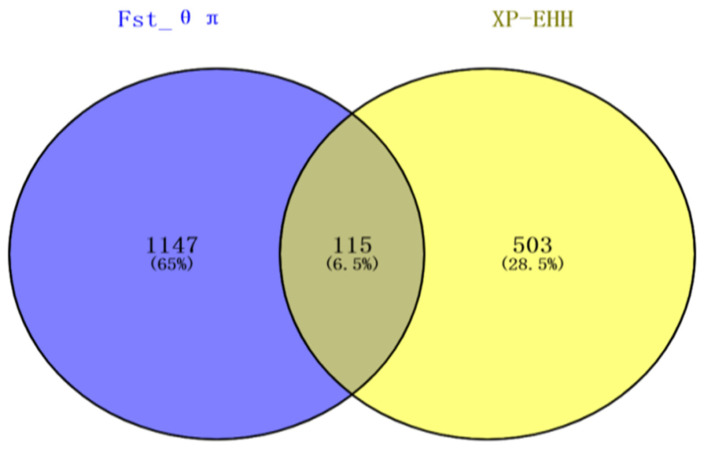
Overlapping SNP of Fst-θπ and XPEHH.

**Table 1 genes-12-01924-t001:** Information of the sheep populations used in this study.

Breed	Abbreviation	Samples Size	Wool Type
Arawapa	APA	37	fine-wool sheep
Australian Poll Merino	APM	98	fine-wool sheep
Brazilian Creole	BCS	23	fine-wool sheep
Chinese Merino	CME	23	fine-wool sheep
Meat Lacaune	LAC	75	fine-wool sheep
Australian Merino	MER	88	fine-wool sheep
Rambouillet	RMB	102	fine-wool sheep
SwissBlack-Brown Mountain Sheep	SBS	24	fine-wool sheep
African Dorper	ADP	21	hair sheep
Adane	AKD	12	hair sheep
Arabo	AKR	10	hair sheep
African White Dorper	AWD	6	hair sheep
Barbados Black Belly	BBB	24	hair sheep
Morada Nova	BMN	22	hair sheep
Bonga	BQ	9	hair sheep
Santa Ines	BSI	47	hair sheep
Doyogena	DA/DH	15	hair sheep
Kefis	FKD	13	hair sheep
Gesses	GGD	11	hair sheep
Hammari	H	11	hair sheep
Kabashi	K	9	hair sheep
Kidor	KO	10	hair sheep
Loya	LA	15	hair sheep
Molale	MZ	15	hair sheep
Red Maasai	RMA	45	hair sheep
Shubi Gemo	SHG	15	hair sheep
Gafera	WA	15	hair sheep

**Table 2 genes-12-01924-t002:** Candidate genes associated with hair follicle development and wool shedding traits in hair sheep.

Chromosome ^1^	Position (bp) ^2^	REF ^3^	ALT ^4^	Gene Name ^5^	Distance(bp) ^6^
5	41339449	G	T	*TCF*3	2996
13	52044015	C	T	*TGM*3	825
13	52055150	G	A	*TGM*3	within
13	53089456	C	T	*SOX*18	26,609
14	48786396	G	A	*PRX*	within

^1^ The *Ovis aries* chromosome number of each significant SNP. ^2^ The positions of each significant SNP associated with wool-shedding traits in the *Ovis aries* Oar_4.0 assembly. ^3^ The base in the reference genome corresponding to the mutation site. ^4^ Observed base change. ^5^ The nearest annotated gene to each significant SNP. ^6^ Designation of SNPs as within a gene or at a distance (bp) from a gene region, according to the *Ovis aries* Oar_4.0 assembly (https://www.ncbi.nlm.nih.gov/assembly/GCF_000298735.2, accessed on 16 September 2021).

## Data Availability

Not applicable.

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
