# Peer review of "Genome-Wide Selective Signatures Reveal Candidate Genes Associated with Hair Follicle Development and Wool Shedding in Sheep"

_genes, 2021, doi:10.3390/genes12121924_

Round 1
Reviewer 1 Report
Authors present a fairly expansive population genetics study to identify genetic loci associated with hair type in sheep. While the scale of the work is admirable, a few concerns about the conclusions of their work must be addressed. In particular, simply linking SNPs to a handful of genes is insufficient to make conclusions about function and additional functional information should be added.
Major Comments:
The results presented in Table 2 must include formalized pathway enrichment analyses. Without them, it is unclear if hair-related genes have more nearby relevant SNPs than other non-hair-related genes. For example, the Genomic Regions Enrichment of Annotations Tool (GREAT) can be used to perform pathway enrichment analyses for noncoding regions. Authors could also use a simple count-based test to compare the proportion of hair-related genes with nearby notable SNPs to the proportion of non-hair-related genes with nearby notable SNPs. Enrichment analysis results could also be supplemented with information about skin and hair follicle open chromatin information, if available from sheep.
Aside from enrichment analyses, more transparency should be used when presenting gene-specific results. Specifically, methods related to gene mapping must be explained more explicitly. In Table 2, How were these SNPs selected to be presented specifically? Are they the most significant, the only ones near hair genes, etc.? If they are the only ones near "hair genes", how were the hair genes selected? For the TCF3 and SOX18 SNPs, the distance from gene to SNP is very large. Especially for SOX18, authors should address why they expect a SNP over 20,000 bases away to be associated with SOX18 in particular.
Line 231: The sentence that includes "PCA was used to allocate them to groups using genotypic markers." This was never stated in the methods and should be included there.
Figure 3: A section of sites on chromosome 6 appear to be shifted upward. Do authors have any explanation for that shift? Just from looking at the figure, it seems either like an error or a very interesting signal, but none of the SNPs addressed in particular (in Table 2) are from that region.
Paragraph starting at line 173: Although shedding is one difference between hair and wool sheep, they also differ in other hair phenotypes. Therefore, genomic elements are not necessarily associated with shedding alone and that should be made clear in the text.
Minor Comments:
Paragraph 1: What are "depilation traits"? From my understanding, hair sheep are simply sheep with hair instead of wool, which decreases the amount of coat upkeep required by farmers. This term should be formally defined very early in the manuscript.
Paragraph 1: Remove the word "obvious" because it isn't clear why or how the described traits are obvious.
Line 44: "typing" should be "phenotyping"
Lines 97-101: The sentence that begins "Here, we selected 27…" is a run-on sentence and should be split into at least 2 separate sentences.
Line 105: Should "State Council of the China" be "State Council of China"?
Table 1: Sheep breed names should use spaces, not camel case; for example BrazilianCreole should be Brazilian Creole
Line 173: Remove the word "obvious"
Figure 1 should show sheep breeds in addition to hair/wool classification. This may help to explain the smaller sub-clusters within the hair/wool groupings.
Reviewer 2 Report
The authors identified several important genes related to hair follicle formation, epidermal differentiation, and hair follicle stem cell development in a total of 795 individuals from 27 breeds of sheep.
Generally, the manuscript is very well written and clearly presented, the introduction is informative. The materisl and methods are clearly described and the results are well presented. The discussion is deep enough. However, I have some sugestions:
1. It is possible to insert in the introduction parta a paragraph about the breeds used in the study?
2. Point out in the part of Results, table 2:
- the amino acid change of the mutations within genes (where appropriate)
-whether the AA substitution change the protein structure.
Reviewer 3 Report
The manuscript genes-1439783, entitled “Genome-wide selective signatures reveal candidate genes associated with hair follicle development and depilation traits in sheep”. This study in sheep is elemental research for understanding the genetic mechanisms of depilation traits using selective signatures to identify genes related to sheep hair follicle development and the control of depilation traits and to understand their functions, which can be used to provide a meaningful reference for further study on the molecular mechanisms of economically important traits in sheep. This manuscript is well written and it is possible to understand all that has been done. But authors should provide the complete gene name for each gene mentioned in the introduction section.
Reviewer 4 Report
The manuscript “Genome-wide Selective Signatures Reveal Candidate Genes Associated with Hair Follicle Development and Depilation Traits in Sheep” aimed to investigate the population structures and genetic differences between sheep with different wool types (haired and fine-wool) to identify candidate genes related to these traits. I have not found previous articles that addressed this objective. The manuscript was well designed, with an adequate number of animals and appropriate statistical approach. Moreover, the text contains a detailed information regarding the analyzes, results and discussion. So, I recommend acceptance after minor revisions indicated below.
Lines 44–82 - These three paragraphs describe the estimated population parameters, which are already well known. Furthermore, these descriptions were presented in other parts of the manuscript. Therefore, this intense description of the parameters does not seem necessary to me. I suggest reducing this part of the introduction.
Line 59 – replaced “others” by “others.”
Line 104 – delete “All the animal work involved in”.
Lines 180-182 – What was this threshold? How was this threshold defined? Include these information in material and methods.
Line 213 – before “26 609 bp” include “in Chr13 at a distance of”
Lines 213-214 replace “48786396 located within PRX” by “48786396 in chr14 located within PRX”
